# Reduced Levels of ABCA1 Transporter Are Responsible for the Cholesterol Efflux Impairment in β-Amyloid-Induced Reactive Astrocytes: Potential Rescue from Biomimetic HDLs

**DOI:** 10.3390/ijms23010102

**Published:** 2021-12-22

**Authors:** Giulia Sierri, Roberta Dal Magro, Barbara Vergani, Biagio Eugenio Leone, Beatrice Formicola, Lorenzo Taiarol, Stefano Fagioli, Marcelo Kravicz, Lucio Tremolizzo, Laura Calabresi, Francesca Re

**Affiliations:** 1BioNanoMedicine Center NANOMIB, School of Medicine and Surgery, University of Milano-Bicocca, 20900 Monza, Italy; g.sierri@campus.unimib.it (G.S.); roberta.dalmagro@unimib.it (R.D.M.); beatrice.formicola@unimib.it (B.F.); l.taiarol@campus.unimib.it (L.T.); stefano.fagioli@unimib.it (S.F.); marcelo.kravicz@unimib.it (M.K.); 2School of Medicine and Surgery, University of Milano-Bicocca, 20900 Monza, Italy; barbara.vergani@unimib.it (B.V.); biagioeugenio.leone@unimib.it (B.E.L.); lucio.tremolizzo@unimib.it (L.T.); 3Department of Pharmacological and Biomolecular Science, Centro Grossi Paoletti, University of Milan, 20133 Milan, Italy; laura.calabresi@unimi.it

**Keywords:** brain cholesterol, Alzheimer disease, HDL, ABCA1, astrocytes, nanoparticles, apoA-I nanodiscs

## Abstract

The cerebral synthesis of cholesterol is mainly handled by astrocytes, which are also responsible for apoproteins’ synthesis and lipoproteins’ assembly required for the cholesterol transport in the brain parenchyma. In Alzheimer disease (AD), these processes are impaired, likely because of the astrogliosis, a process characterized by morphological and functional changes in astrocytes. Several ATP-binding cassette transporters expressed by brain cells are involved in the formation of nascent discoidal lipoproteins, but the effect of beta-amyloid (Aβ) assemblies on this process is not fully understood. In this study, we investigated how of Aβ_1-42_-induced astrogliosis affects the metabolism of cholesterol in vitro. We detected an impairment in the cholesterol efflux of reactive astrocytes attributable to reduced levels of ABCA1 transporters that could explain the decreased lipoproteins’ levels detected in AD patients. To approach this issue, we designed biomimetic HDLs and evaluated their performance as cholesterol acceptors. The results demonstrated the ability of apoA-I nanodiscs to cross the blood–brain barrier in vitro and to promote the cholesterol efflux from astrocytes, making them suitable as a potential supportive treatment for AD to compensate the depletion of cerebral HDLs.

## 1. Introduction

Cholesterol is an essential component of cell membranes and it is involved in numerous biological functions. Brain, despite accounting for only 2% of the body weight, contains about 25% of total cholesterol [1]. The Central Nervous System (CNS) depends almost entirely on the endogenous synthesis of cholesterol because of the presence of the Blood–Brain Barrier (BBB), which limits the exchange of molecules between the blood and brain compartment [2].

The cholesterol synthesis, in adulthood, is mainly handled by astrocytes, which are also responsible for apoproteins’ synthesis and lipoproteins’ assembly required for the cholesterol transport in the brain parenchyma [3]. The mechanisms that lead to the formation of nascent lipoproteins in the brain are not fully understood. Several ATP-binding cassette (ABC) transporters, such as ABCA1 and ABCG1, are expressed in the CNS and they are involved in the regulation of cholesterol efflux from cells to nascent lipoproteins. ABCA1 mediates the efflux of cholesterol from astrocytes to apolipoproteins, apoE and apoJ, leading to the formation of discoidal lipoproteins. The ABCG1 and ABCG4 transporters expressed on the astrocyte membrane mediate further incorporation of cholesterol and phospholipids into nascent discoidal lipoproteins.

A small fraction of cholesterol can be also taken up from the blood circulation as 27-hydroxycholesterol or via lipoprotein receptors (i.e., the scavenger receptor class B type I, (SR-B1) or low-density lipoproteins receptor (LDLR)) that are responsible for the cerebral uptake of peripheral lipoproteins [4].

Cholesterol levels and turnover are impaired in neurodegenerative disorders, and the cholesterol transport and recycling in the brain seem to play a key role in the development and progression of such diseases [5,6]. Among them, Alzheimer disease (AD) represents the most frequent form of dementia affecting the elderly population. AD is histologically characterized by the accumulation of two different proteins: the β-amyloid protein (Aβ), which is deposited extracellularly, and the hyperphosphorylated Tau protein (pTau), which constitutes intracellular neurofibrillary tangles. Another feature of AD is the astrocytic reactivity, or astrogliosis, a non-transient process involving astrocytes. Astrogliosis is accompanied by functional and morphological changes of astrocytes with a significant upregulation of glial fibrillary acidic protein (GFAP) [7]. Experiments on AD-like transgenic mice have demonstrated that silencing ABCA1 decreases the lipidation of apoE, increases the deposition of Aβ, and reduces drastically high-density lipoproteins’ (HDLs) levels in the brain parenchyma and in the cerebrospinal fluid (CSF) [8,9]. This likely occurs because lipid-poor apoE promotes Aβ aggregation and toxicity, reducing its clearance [10]. ApoA-I is the second most abundant lipoprotein after apoE in the CSF [11] Since apoA-I mRNA has not been detected in the brain [12], its presence is thought to be delivered from the periphery through the BBB. Several studies have shown that levels of HDLs and apoA-I are reduced in AD patients compared to healthy controls, in blood circulation [13].

Starting from these premises, the aim of this work was to define whether defects in brain cholesterol transporters at the cellular level contribute to the alteration of cholesterol transport in an AD-like condition. We detected an impairment in the cholesterol efflux of reactive astrocytes attributable to reduced levels of ABCA1 transporter. These results could contribute to explain the reduced lipoproteins’ levels detected in AD patients [9,13]. In addition, we designed biomimetic HDLs to test the hypothesis that the modulation of cerebral lipoprotein-mediated cholesterol transport could be a valuable strategy for AD treatment.

## 2. Results and Discussion

### 2.1. Assessment of Aβ_1-42_ Aggregation

Normal human astrocytes (NHA) were treated with different aggregation forms of Aβ_1-42_ peptide, the most enriched Aβ peptide in amyloid plaques [14], to mimic the astrocytic reactivity detected in AD. The formation of oligomers or fibrils was checked by SDS-PAGE/WB. The results showed that oligomers’ preparations was enriched with soluble Aβ species with MW up to 40 kDa without large-sized aggregates, as indicated by the absence of smears at high molecular weights. Fibrils-enriched samples were characterized by smears at MW > 40 kDa, accompanied by a reduction in bands of small-sized aggregates, likely corresponding to an enrichment in large-molecular-weight aggregates (Figure 1a,b). These results were consistent with those present in literature [15,16]. Aβ_1-42_ aggregation was also monitored by measuring the turbidity of solutions at 330 nm and by thioflavin T (ThT) binding fluorescence assay. The results showed that the turbidity of solution and the ThT fluorescence intensity progressively increased over time, suggesting the formation of amyloid-like species (Figure 1c,d), according to previous results [17].

### 2.2. Assessment of Cell Viability after Exposure to Aβ

To set the dosage of Aβ_1-42_ at which NHA cell viability was preserved and astrogliosis induced, MTT cytotoxicity assay was performed. NHA were treated with different doses of oligomers or fibrils for 24 or 48 h and the cells’ viability was measured by MTT assay.

The results showed that the cell viability was ≥50% for all the dose and time conditions tested without significant differences between oligomers and fibrils. The NHA viability was reduced <50% only when treated with the highest dose of oligomers for 48 h (Figure 2).

The higher toxicity of oligomers with respect to fibrils was in agreement with data present in literature, showing that the smaller soluble oligomers are the most toxic species of Aβ [18], probably because of their higher cellular uptake compared to large aggregates [19].

### 2.3. Evaluation of Aβ_1-42_-Induced Astrogliosis

To evaluate the Aβ-induced astrogliosis, GFAP levels were measured in NHA treated with different concentrations of oligomers or fibrils for 24 or 48 h by SDS-PAGE/WB.

GFAP showed a characteristic electrophoretic pattern in which different isoforms were visible with molecular weights variable between 35 and 50 kDa, which corresponded to different post-translational modifications of the protein. The isoform of approximately 45 kDa was the most representative one [20]. Results showed that the treatment with Aβ_1-42_ oligomers induced a dose-dependent increase of the GFAP expression levels, both after 24 (Figure 3a) and 48 h of incubation (Figure 3b), suggesting the progressive increase of astrocytes’ reactivity [7]. It was possible to observe that the increase of GFAP levels was more evident after 24 h of incubation, with respect to the levels at 48 h at the same doses. This difference could be due to a change of oligomers/fibrils ratio in the samples, with a relative increase of fibrils and decrease of oligomers, which anyway represented the largest portion of Aβ.

On the contrary, the presence of Aβ_1-42_ fibrils caused a reduction of GFAP expression levels. GFAP levels were halved after incubation with Aβ_1-42_ concentration ≥10 μM after 24 h of incubation (Figure 4a). After 48 h of Aβ_1-42_ treatment, the reduction of GFAP was dose dependent and GFAP levels were halved with 20 μM Aβ_1-42_ (Figure 4b)

A reduction of GFAP expression levels has been observed in some pathological conditions, such as meningiomas and neurinomas [21,22], showing that perturbation of the astrocytes’ state could differently affect the expression levels of this protein. The result obtained will deserve further investigation for a better understanding of this alteration in the presence of Aβ fibrils. Overall, these results showed that only the Aβ_1-42_ oligomer-enriched preparation induced an increase of GFAP levels, making astrocytes reactive. To the best of our knowledge, the different astrocytic response to oligomers or fibrils has not yet been reported.

Based on MTT and GFAP levels’ results, the subsequent studies were conducted by using oligomer-enriched samples at two concentrations, 1 and 10 µM, of Aβ_1-42_ to mimic an AD-like condition.

### 2.4. Evaluation of Cholesterol Efflux

The cholesterol efflux was evaluated on NHA treated with oligomers for 24 or 48 h, using ACM (astrocytes’ conditioned medium), containing apolipoprotein E (Appendix A) as a cholesterol acceptor. Results showed that the cholesterol efflux decreased, reaching a reduction of 25% after 24 h of treatment with Aβ_1-42_ at both concentrations tested, in comparison to untreated NHA (Figure 5a). By increasing the time of treatment to 48 h, a −35% decrease of the cholesterol efflux was measured at the highest Aβ_1-42_ dose (Figure 5b). Considering the metabolic dependency of neurons on cholesterol produced and released by glial cells [23], an impairment of astrocytic cholesterol efflux may affect the neuronal activities.

### 2.5. Evaluation of ABCA1 and ABCG1 Levels in Reactive NHA

In order to explain this reduction, the expression levels of the membrane transporters ABCA1 and ABCG1, which are involved in the regulation of cholesterol efflux from cells, were evaluated. NHA were treated with Aβ_1-42_ oligomers at different doses and times, and the levels of transporters were measured by SDS-PAGE/WB. Results showed that, after a 48 h of treatment, a significant reduction (−40%) of ABCA1 expression was detected for both the Aβ_1-42_ concentrations tested. A stronger reduction (−55%) was observed after 72 h of treatment with the highest Aβ_1-42_ dose (Figure 6a), suggesting that the effect of oligomers is persistent over time. On the contrary, no significant changes in ABCG1 levels were detected after Aβ treatment (Figure 6b). This reduction in ABCA1 levels may justify the reduction in cholesterol efflux detected after treatment with Aβ.

It is noteworthy that 24 h of treatment with oligomers induced an initial increase of ABCA1 levels in NHA cells, which was not accompanied by an increase of cholesterol efflux rate. This discrepancy may be imputable to a functional alteration of ABCA1 transporter or it can be interpreted as an effort to clear Aβ through the modulation of membrane cholesterol efflux [24]. The mechanism by which Aβ acts on ABCA1 level and activity and the role of ABCA1 in Aβ clearance deserves further investigation.

Considering that cerebral ABCA1 is involved in the lipidation of apoproteins, such as apoE, apoA-I, and apoJ to generate discoidal HDL-like lipoproteins, the reduction of this transporter could affect the cerebral HDL levels. Accordingly, it has been shown that silencing cerebral ABCA1 transporter in mouse models induces a substantial decrease of lipoproteins’ levels in the brain [25], thus affecting the cholesterol transport.

### 2.6. Characterization of Biomimetic Nanodiscs

To this purpose, in order to elude this issue, we designed apoA-I nanodiscs to mimic cerebral HDLs with the aim to restore cerebral cholesterol transport. Although the cholesterol delivery in the brain is a primary function of apoE, it is also well known that apoE is involved in Aβ aggregation [10]. For this reason, we synthetized lipidic nanodiscs composed of apoA-I, apoprotein that holds neuroprotective properties, such as inhibition of Aβ aggregation [26].

Biomimetic HDLs were characterized by Dynamic Light Scattering (DLS) and Transmission electron microscopy (TEM). The final preparations of nanodiscs were monodispersed (PDI = 0.2) with a size <95 nm (Table 1), in line with already published data [27,28].

TEM imaging showed the presence of flat, circular, disk-shaped structures (Figure 7a). The ellipsoid shape in the background represents face-on or front side of the nanodiscs, while elongated shapes represent their flank side, depending on their orientation. The internal structure of the nanodiscs was not clearly discernible by TEM. However, striations visible in the enlargement (Figure 7b) might represent apoA-I wrapped around a central ellipsoidal lipid phase, as shown for HDLs isolated from human plasma [28].

### 2.7. Evaluation of apoA-I Nanodiscs as Cholesterol Acceptors

To evaluate the ability of the nanodiscs to mimic the activity of human discoidal HDLs, their performance in promoting the cholesterol efflux from the NHA was assessed. Results demonstrated that the use of apoA-I nanodiscs promoted a 3-fold increase (*p* = 0.0002) of the cholesterol efflux from NHA cells, compared to the control. Moreover, their capability in enhancing cholesterol efflux was comparable to that of ACM (Figure 8), suggesting that their activity is similar to apoE-lipoproteins produced by astrocytes.

To evaluate the performance of apoA-I nanodiscs in restoring the impaired cholesterol efflux, NHA cells were treated with Aβ_1-42_ oligomers at 1 μM or 10 μM for 24 h and nanodiscs were used as cholesterol acceptor in comparison to ACM. The cholesterol efflux from the cells was measured as above. Results demonstrated that apoA-I nanodiscs promoted an increase of cholesterol efflux from NHA cells treated with Aβ_1-42_ oligomers, suggesting their efficacy in rescuing the cholesterol homeostasis. However, it was possible to notice that the efficacy of apoA-I nanodiscs in boosting the cholesterol efflux from treated cells was reduced with the increasing of Aβ dose (Figure 9a). Therefore, an optimization of dose and time of incubation with nanodiscs will be defined. In order to understand if the restored cholesterol efflux in the presence of apoA-I nanodiscs improves also cellular processes related to AD, which are compromised of oligomers, cell viability, GFAP levels, and ABCA1 levels were measured after 4 h of incubation with apoA-I nanodiscs of Aβ-treated NHA cells. The results showed that the nanodiscs’ treatment induced the increase of ABCA1 levels, explaining the increased cholesterol efflux. Moreover, a reduction of GFAP levels was also detected, suggesting a nanodiscs-dependent reduction of astrogliosis. Unfortunately, no changes in the cell viability were observed (Figure 9b).

### 2.8. Assessment of Biomimetic Nanodiscs’ Ability to Cross the BBB

In sight of these results, we proposed the utilization of biomimetic HDLs to compensate the depletion of cerebral discoidal HDLs consequent to the reduction of ABCA1 levels induced by Aβ oligomers. In this context, the ability of nanodiscs to cross the BBB was evaluated in a transwell in vitro model using hCMEC/D3 cells. This system allowed the measurement of the passage of nanoparticles across an endothelial monolayer resembling the main features of the BBB. The electrical and functional properties of this model were determined by measuring the Transendothelial electrical resistance (TEER) and the endothelial permeability (PE) with a fluorescent probe, respectively. The results showed that TEER values progressively increased over time, reaching the maximum value between 40–45 Ω × cm^2^ starting from the fifth day after cell seeding (Figure 10a). These values were in agreement with those reported in literature for the same cell line [29]. To evaluate the formation of tight junctions, which are involved in the restriction of paracellular passage of large hydrophilic molecules, the PE to TRITC-dextran (4400 Da) was measured. The results showed that the PE of the fluorescent probe across the endothelial monolayer was 9.54 ± 0.86 × 10^−5^ cm/min, similar to already published data [30].

The increase of TEER values and the restricted permeability to TRICT-dextran confirmed the presence of functional junctions between the endothelial cells.

Fluorescent nanodiscs were added to the apical compartment of the transwell, and the fluorescence in the basolateral compartment was measured over time up to 3 h.

Results demonstrated that apoA-I nanodiscs showed a better ability to cross the BBB, with a 2-fold higher PE value compared to that of nanodiscs without the protein (Figure 10b).

The measured PE values were in the same order of magnitude of those already reported for human discoidal HDLs [31], even if lower (1.5 × 10^−5^ cm/min for apoA-I nanodiscs vs. 13.5 × 10^−5^ cm/min for discoidal HDL). This discrepancy was probably due to the different size of HDL particles (~90 nm for apoA-I nanodiscs vs. ~10 nm).

The passage of apoA-I nanodiscs across the BBB was probably allowed by the presence of the scavenger receptor class B type 1 (SR-B1) receptor on the surface of the hCMEC/D3 (Appendix A), which is responsible for the transcytosis of HDLs and apoA-I through the BBB in vivo [32,33].

Overall, based on these results, we speculated that Aβ oligomers might affect the cholesterol efflux from astrocytes by inducing astrocytic reactivity and by reducing the levels of ABCA1 transporters. These events may explain the reduced levels of cerebral lipoproteins detected in AD patients [9], which in turn can negatively affect neurons’ and other brain cells’ healthiness. Moreover, levels and lipidation state of brain apoproteins may affect the progression of the disease [24].

The design of apoA-I nanodiscs, which are capable to cross in vitro the BBB and to mimic the structure and function of cerebral discoidal HDLs, could represent a possible strategy to compensate the altered cholesterol transport in AD.

## 3. Materials and Methods

### 3.1. Cell Lines

The NHA (Normal Human Astrocytes, Lonza) were cultured in Astrocytes’ Basal Medium (ABM), supplemented with 3% fetal bovine serum (FBS), 1% L-Glutamine, 0.1% gentamicin sulfate/amphotericin (GA-1000), 0.1% ascorbic acid, 0.1% human epidermal growth factor (hEGF), and 0.25% insulin. The medium was changed every day and the cells were used in up to 10 population doublings.

Immortalized human cerebral microvascular endothelial cells (hCMEC/D3) were provided by Dr. S. Bourdoulous (Institut Cochin, Inserm, Paris, France) and used as a model of the brain capillary endothelium (Weksler B et al., 2013). Cells at passage between 25 and 35 were seeded on tissue culture flasks, pre-treated with rat tail collagen type I (0.05 mg/mL). Cells were grown in complete culture medium (EBM-2 supplemented with 10% FBS, 1% Chemically Defined Lipid Concentrate (CDLC), 1% penicillin/streptomycin (P/S), 10 mM Hepes, 5 µg/mL ascorbic acid, 1 ng/mL bFGF, and 1.4 µM hydrocortisone) and maintained at 37 °C, 5% CO_2_. The culture medium was changed every 2 days [34].

### 3.2. Preparation and Characterization of Aβ_1-42_ Aggregates

The peptide Aβ_1-42_ (AnaSpec, Milano, Italy) was solubilized in 1,1,3,3,3-hexafluoro-2-propanol (HFIP; Sigma–Aldrich, Milano, Italy) (1 mg/mL) and air-dried in a chemical fume hood overnight.

Peptide was resuspended in DMSO (5 mM) and bath sonicated for 10 min to obtain a solution enriched in monomers. To obtain an oligomer-enriched preparation, the peptide was diluted in unsupplemented ABM to 100 μM and used immediately. To obtain a fibril-enriched preparation, the peptide was diluted in unsupplemented ABM to 100 μM and incubated at 37 °C for 24 h [35]. The amount of Aβ_1-42_ in the samples was measured by Bradford assay [36].

The aggregation state of the samples was evaluated by SDS-PAGE gel electrophoresis on a 4–12% Bis-Tris-Glycine gel (Thermo Fisher Scientific, Milano, Italy), followed by immunoblotting analysis using 6E10 anti-Aβ antibody (1:1000, Signet). Aβ assemblies were visualized with an enhanced chemiluminescence system by Amersham Imager 600 (GE Healthcare Srl, Milano, Italy). The aggregation over time was monitored by ThT fluorescence assay, which identifies amyloid-containing β-sheet structures, as described in [17], with minor modifications. Briefly, Aβ_1-42_ was added with 10 μM ThT directly in Costar 96-well black plates. The change in ThT fluorescence was monitored continuously for a period of time at 37°C under agitation using fluorescence spectrofluorometer (Varian, Carry Eclipse) at excitation of 445 nm. The formation of aggregates was also monitored by measuring the turbidity of solutions at 330 and 400 nm by UV spectrophotometer in a 96-well UV-transparent plate [37].

### 3.3. In Vitro Cellular Model of Astrogliosis

NHA cells were plated in a 96-well plate (15,000 cells/cm^2^) or Petri dishes (250,000 cells/cm^2^) and incubated with increasing doses of Aβ_1-42_ oligomers or fibrils (ranging from 1–20 μM) in complete medium without FBS, for 24 or 48 h. At the end of the incubation, the cell viability was evaluated by MTT assay as described [18].

The Aβ_1-42_ -induced astrogliosis was verified by measuring the GFAP levels. Briefly, after incubation cells were lysed in Radioimmunoprecipitation assay (RIPA) lysis buffer (25 mM Tris-HCl pH 7.6, 150 mM NaCl, 1% NP-40, 1% sodium deoxycholate, 0.1% SDS) following the manufacturer’s instructions (Thermo Fisher Scientific, Milano, Italy). An aliquot of cell lysate containing 10 μg of total proteins was loaded into gel pre-cast 4-12% Bis-Tris followed by immunoblotting analysis using anti-GFAP antibody (1:1500, Dako, Milano, Italy). GFAP bands were visualized by an enhanced chemiluminescence system using Amersham Imager 600 (GE Healthcare Srl, Milano, Italy). All the data were normalized to β-actin (anti-β-actin 1:5000, Thermo Fisher Scientific, Milano, Italy).

### 3.4. Cholesterol Efflux Assay

To evaluate changes in cholesterol efflux in the presence of Aβ_1-42_, NHA cells were plated in 96-well plate (15,000 cells/cm^2^) and incubated with different concentrations of oligomers (1 μM and 10 μM) for 24 or 48 h in an unsupplemented ABM medium. Cholesterol efflux was measured using the Cholesterol Efflux Assay kit (Sigma Aldrich, Milano, Italy), following the manufacturer’s instructions, using Astrocytes’ Conditioned Medium (ACM) as a cholesterol acceptor.

ACM was obtained by collecting the astrocyte culture medium after 48 h of growth. ACM was filtered through a 0.22-μm filter, and the presence of apolipoprotein E (apoE), as cholesterol acceptor, was evaluated by SDS-PAGE gel electrophoresis on a 4–12% Bis-Tris-Glycine gel (Thermo Fisher Scientific, Milano, Italy), followed by immunoblotting analysis using anti-apoE antibody (1:100, Santa Cruz Biotechnology). ApoE bands were visualized by an enhanced chemiluminescence system by Amersham Imager 600 (GE Healthcare Srl, Milano, Italy). The percentage of cholesterol efflux in the samples was calculated according to the formula: C = [Fm/(Fm + Fc)] × 100% where Fm is the fluorescence intensity of the supernatants and Fc is the fluorescence intensity of the cell lysate. Each value of Fm and Fc was corrected for white.

### 3.5. Evaluation of ABCA1 and ABCG1 Levels in Reactive NHA

NHA cells were plated in Petri dishes (250,000 cells/cm^2^) and incubated with increasing concentrations of Aβ_1-42_ oligomers (ranging from 1–10 μM) in complete ABM medium without FBS for 24, 48, or 72 h to evaluate the expression of ABCA1 and ABCG1 transporters.

Briefly, after incubation cells were lysed in RIPA buffer (Thermo Fisher Scientific, Milano, Italy) following the manufacturer’s instructions. An aliquot of cell lysate containing 10 μg of total proteins was loaded into gel pre-cast 3-8% Tris-acetate followed by immunoblotting analysis using anti-ABCA1 (1:500, Thermo Fisher Scientific, Milano, Italy) and anti-ABCG1 antibody (1:1000, Thermo Fisher Scientific). Bands were visualized by an enhanced chemiluminescence system using Amersham Imager 600 (GE Healthcare Srl, Milano, Italy). All the data were normalized to β-actin (anti-β-actin 1:5000, Thermo Fisher Scientific, Milano, Italy).

### 3.6. Preparation and Characterization of Biomimetic Discoidal HDLs

Nanodiscs were prepared following the procedure described by Dai T. et al. [28]. Briefly, 1-palmitoyl-2-oleoyl-sn-glycero-3-phosphocholine, cholesterol, and 1,2-distearoyl-sn-glycero-3-phosphoethanolamine-N-[amino(polyethylene glycol)-2000] (Sigma Aldrich, Milano, Italy) were mixed in a molar ratio of 35:40:25 (POPC:Chol:PEG-DSPE) and rotary-evaporated to form a thin film by removing solvent. The lipid film was dried under vacuum overnight and hydrated in phosphate-buffered saline (PBS) for 1 h at 37 °C. Nanodiscs were subsequently prepared by sonication of the hydrated solution for 45 min in an ice-bath using a Sonics Vibra-Cell (Pasquali Ettore S.R.L., Milano, Italy). The resulting solution was filtered through a 0.22-μm filter to remove metal debris.

Apo A-I nanodiscs were prepared using POPC:Chol:apoA-I in the molar ratio of 100:10:1 [28]. Lipids were dissolved in chloroform/methanol (2:1, *v*/*v*), dried under vacuum. and the lipid film was then rehydrated with human plasma-derived apoA-I (Sigma Aldrich, Milano, Italy) dissolved in PBS pH 7.4 (1.3 mg/mL) for 1 h at 37 °C. ApoA-I nanodiscs were prepared by sonication, as described above. Free apoA-I protein was removed by dialysis (Spectra/Por Dialysis Membrane MWCO: 50,000) against PBS overnight at 4 °C.

For BBB permeability assay, nanodiscs were labelled by adding 1 mol% PE-Rhodamine B (Sigma-Aldrich, Milano, Italy) to the lipid mixture.

Nanodiscs were characterized in terms of lipid recovery by Stewart’s assay [38], size by dynamic light scattering (DLS) [39], and morphology through the negative staining technique using a Philips CM10 transmission electron microscope (TEM).

### 3.7. Functional Characterization of Nanodiscs as Cholesterol Acceptors

The ability of nanodiscs to mimic the function of endogenous discoidal HDLs in promoting cholesterol efflux was evaluated, as described above on NHA cells, untreated or treated with Aβ_1-42_ oligomers (ranging from 1–10 μM) for 24 h. Briefly, untreated NHA cells were incubated with ACM, free apoA-I (20 μg/mL), apoA-I nanodiscs (20 μg/mL apoA- I, 0.05 μmol lipids), or nanodiscs (0.05 μmol lipids). NHA cells treated with Aβ were incubated with ACM or apoA-I nanodiscs (40 μg/mL apoA- I), both at 37 °C for 4 h. Cholesterol efflux was measured using the Cholesterol Efflux Assay kit (Sigma Aldrich, Milano, Italy), following the manufacturer’s instructions. Cell viability, ABCA1, and GFAP levels were also measured, as above described.

### 3.8. Preparation and Characterization of the in Vitro Model of Blood–Brain Barrier

The in vitro BBB model was prepared and characterized, as previously described [35], using hCMEC/D3 cells. Briefly, cells were seeded (56,000 cells/cm^2^) onto collagen-coated (150 μg/mL rat tail collagen type 1; Gibco, Thermo Fisher Scientific) transwell filters (polyester 12-well, pore size 0.4 µm, translucent membrane inserts 1.12 cm^2^; Costar) to establish a polarized monolayer. The cell monolayer separates into two compartments, an apical one (0.5 mL) representing the blood and a basolateral one (1 mL) representing the brain. Cells were grown for 3 days in complete EBM-2 medium. After 3 days, the medium was replaced with EBM-2 supplemented with 5% FBS, 1% CDLC, 1% P/S, 10 mM Hepes, 5 μg/mL ascorbic acid, 1.4 μM hydrocortisone, and 10 mM LiCl. The formation of junctions was evaluated by measuring TEER, monitored with STX2 electrode Epithelial Volt-Ohm meter (World Precision Instruments, Sarasota, FL, United States) and the paracellular permeability to TRITC-dextran 4400 Da (λecc = 557 nm, λem = 572 nm) (Sigma Aldrich, Milano, Italy) [40].

The SR-B1 expression by hCMEC/D3 was evaluated by SDS-PAGE/WB. Briefly, cells were lysed in RIPA buffer (ThermoFisher Scientific, Milano, Italy). An aliquot of cell lysate containing 30 μg of total proteins was loaded onto Bolt 4–12% Bis-Tris Plus gel (ThermoFisher Scientific, Milano, Italy), followed by immunoblotting analysis using an anti-SR-B1 antibody (1:2000, Abcam, Milano, Italy). SR-B1 bands were visualized by an enhanced chemiluminescence system using Amersham Imager 600 (GE Healthcare Srl, Milano, Italy).

### 3.9. Assessment of Biomimetic HDLs’ Ability to Cross the BBB In Vitro

On the seventh day of the hCMEC/D3 culture, at the highest TEER values and lowest PE of TRITC-dextran, fluorescent nanodiscs or apoA-I nanodiscs were added to the apical compartment (0.7 μmol of lipids, 0.007 μmol apoA-I) and then incubated at 37 °C for up to 3 h. At different time points, the PE-Rhodamine B fluorescence in the basolateral compartment was measured at λecc = 560 nm and λem = 590 nm using a spectrofluorometer (Spectrofluorometer FP-8500, Jasco).

### 3.10. Statistical Analysis

Each experiment was conducted at least in triplicate and the data were expressed as mean ± standard deviation or SEM. Statistical differences were evaluated by two-way ANOVA Tukey’s multiple comparisons test, one-way ANOVA, and Student’s *t* test. *p*-values < 0.05 were considered statistically significant.

## Figures and Tables

**Figure 1 ijms-23-00102-f001:**
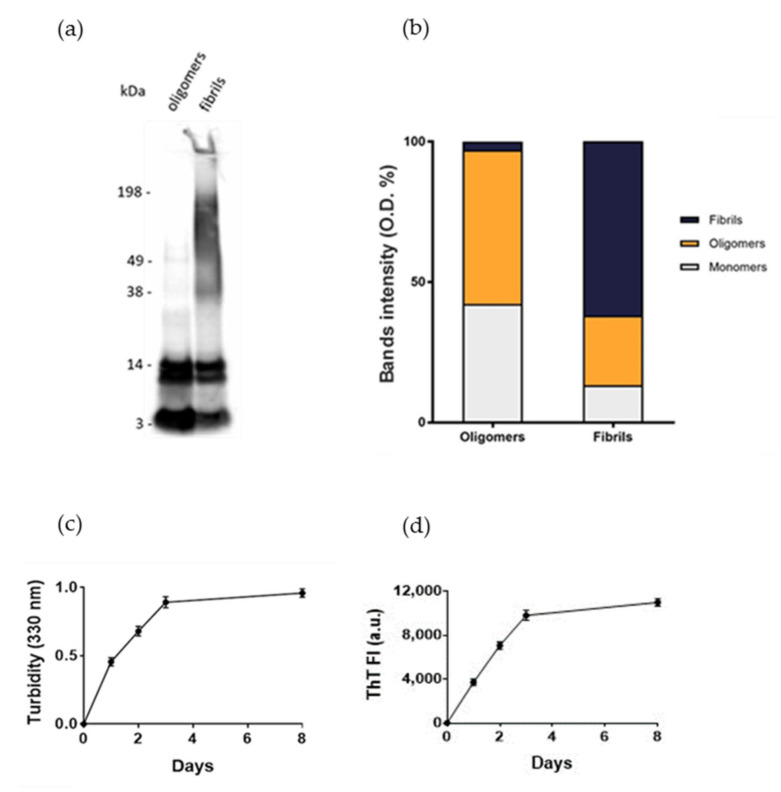
Assessment of Aβ aggregation. (**a**) Oligomers’ and fibrils’ preparations of Aβ_1-42_ were characterized by SDS-PAGE/WB. Representative Western blot of Aβ_1-42_ probed with anti-Aβ 6E10 and visualized by ECL is shown. (**b**) Quantification of the optical density (O.D.) of bands, expressed as %. (**c**) Turbidity at 330 nm. (**d**) Thioflavin T (ThT) fluorescence. Experiments were performed in triplicate and data are presented as mean ± standard deviation.

**Figure 2 ijms-23-00102-f002:**
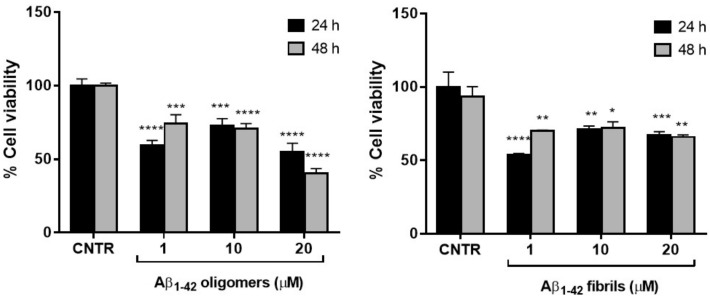
Cell viability after exposure to Aβ. NHA cells were incubated with different doses of Aβ oligomers (**a**) or fibrils (**b**) ranging from 1 to 20 μM, for 24 or 48 h. The cytotoxicity of treatments was assessed by MTT assay; * *p* < 0.05; ** *p* < 0.01; *** *p* < 0.001; and **** *p*< 0.0001 (two-way ANOVA).

**Figure 3 ijms-23-00102-f003:**
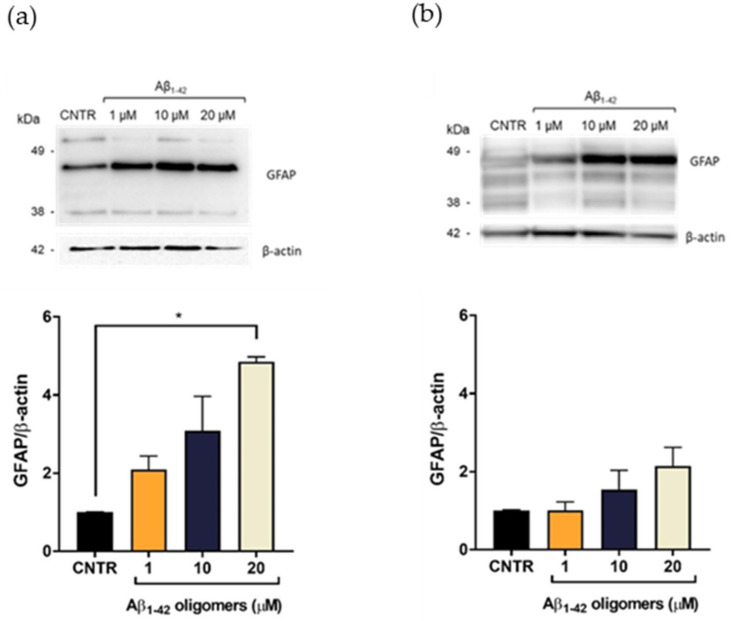
Quantification of GFAP expression levels in NHA after exposure to Aβ_1-42_ oligomers via SDS-PAGE/WB. NHA cells were treated with a preparation of Aβ_1-42_, enriched in oligomers, at increasing concentrations (1 μM, 10 μM, and 20 μM) for 24 (**a**) and 48 (**b**) h. The data are expressed as the mean of the samples in triplicate ± SEM; * *p* <0.05 (one-way ANOVA).

**Figure 4 ijms-23-00102-f004:**
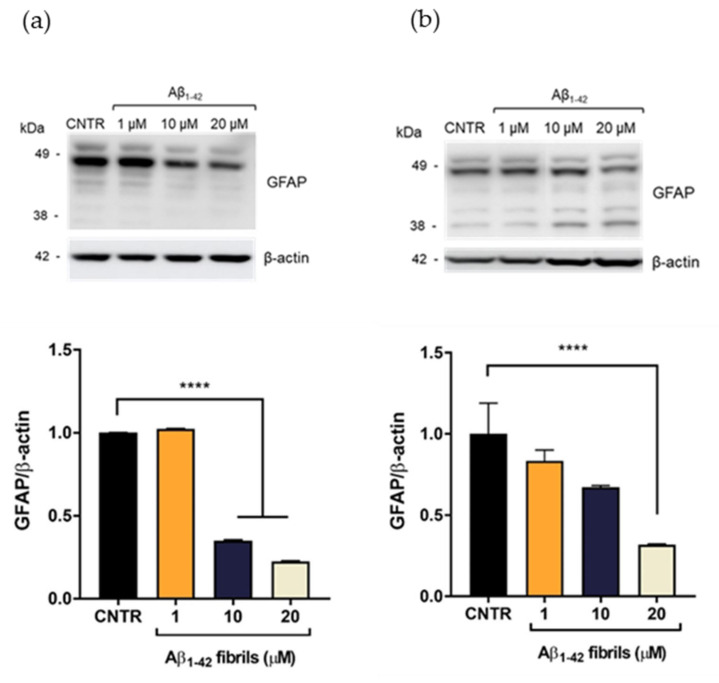
Quantification of GFAP expression levels in NHA after exposure to Aβ_1-42_ fibrils via SDS-PAGE/WB. NHA cells were treated with a preparation of Aβ_1-42_, enriched in fibrils, at increasing concentrations (1 μM, 10 μM, and 20 μM) for 24 (**a**) and 48 (**b**) h. The data are expressed as the mean of the samples in triplicate ± SEM; **** *p* < 0.0001 (one-way ANOVA).

**Figure 5 ijms-23-00102-f005:**
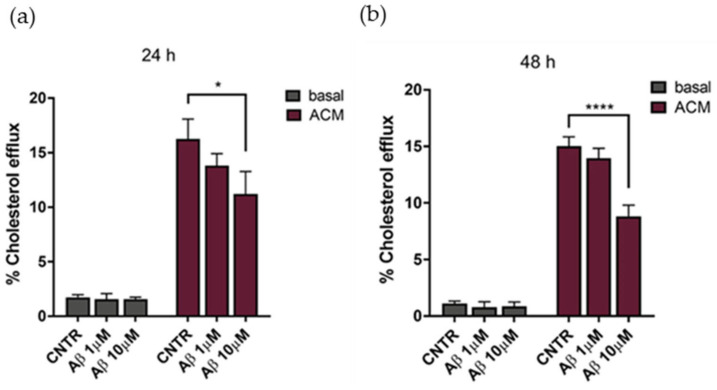
Evaluation of cholesterol efflux from astrocytes in AD-like conditions, in the presence of ACM as acceptor of cholesterol. The cholesterol efflux was measured using the Cholesterol Efflux Assay kit, after treatment with Aβ_1-42_ oligomers (1 μM or 10 μM) for 24 (**a**) or 48 (**b**) h. The fluorescence of supernatant and of cellular lysates was measured by spectrofluorometer. Data are expressed as the mean of the samples in triplicate ± SEM; * *p* < 0.05; **** *p* < 0.0001 (2way ANOVA).

**Figure 6 ijms-23-00102-f006:**
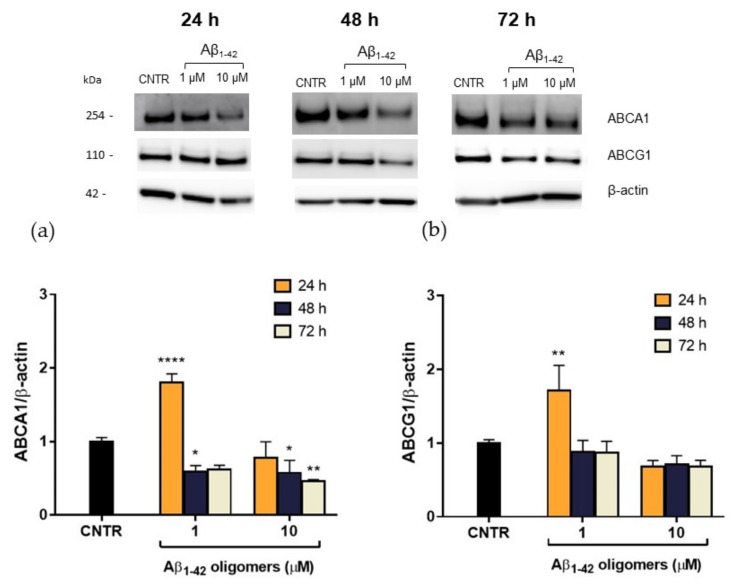
Quantification of ABCA1/ABCG1 expression levels in NHA after exposure to Aβ_1-42_ oligomers via SDS-PAGE/WB. NHA cells were treated with increasing concentrations (1 μM, 10 μM) of oligomers-enriched Aβ_1-42_ samples for 24, 48, or 72 h, and ABCA1 (**a**) and ABCG1 (**b**) levels were evaluated. The data are expressed as the mean of the samples in triplicate ± SEM; * *p* < 0.05; ** *p* < 0.01; **** *p* < 0.0001 (two-way ANOVA).

**Figure 7 ijms-23-00102-f007:**
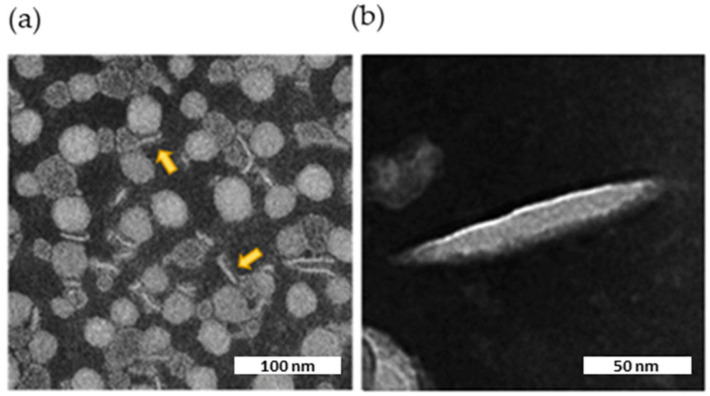
TEM image of the nanodiscs. Characteristic negative stain electron microscopic image of nascent HDL, along with two expanded views of the indicated insets. The scale bar corresponds to 100 or 50 nm. (**a**) Nanodiscs viewed laterally are indicated by yellow arrows. (**b**) Enlargement of a nanodisc decorated with apoA-I.

**Figure 8 ijms-23-00102-f008:**
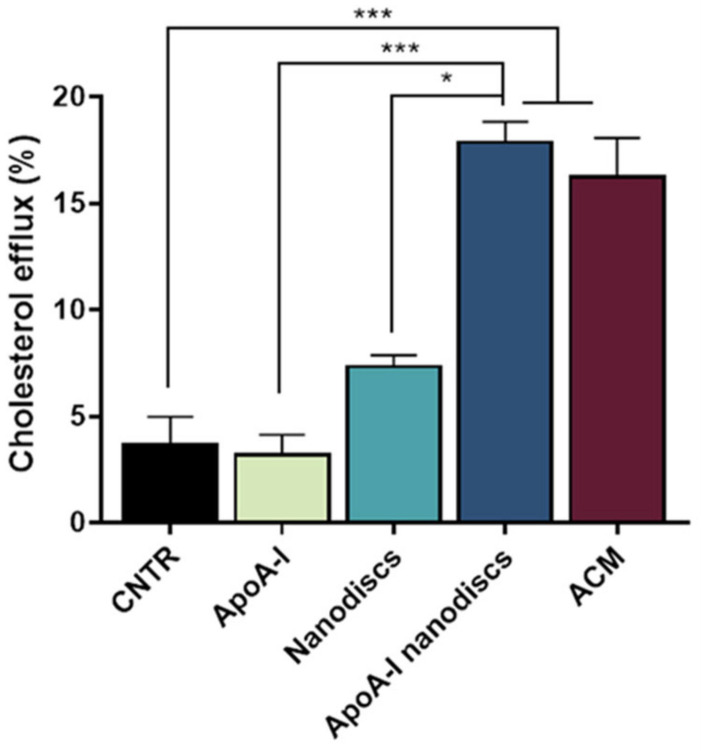
Evaluation of the ability of apoA-1 nanodiscs as cholesterol acceptors. The cholesterol efflux was measured using the Cholesterol Efflux Assay kit. NHA cells were treated with different cholesterol acceptors: free apoA-I, nanodiscs, apoA-I nanodiscs, and ACM. The fluorescence of supernatant and of cell lysates was measured by spectrofluorometer; * *p* < 0.05; *** *p* < 0.001 (one-way ANOVA).

**Figure 9 ijms-23-00102-f009:**
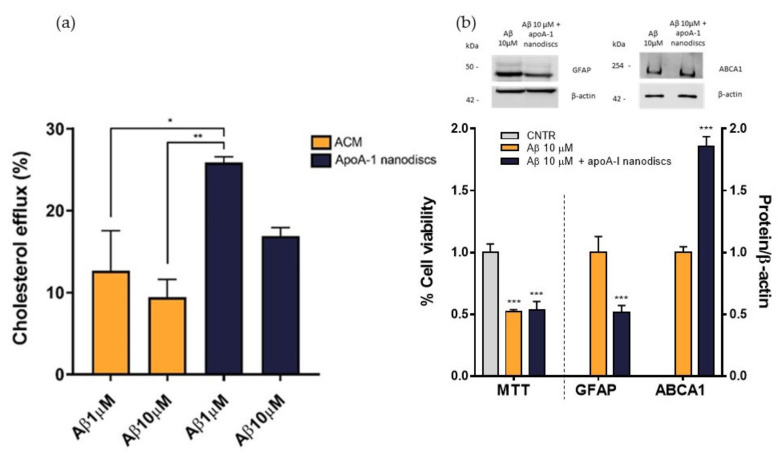
Evaluation of apoA-I nanodiscs efficacy in AD-like condition. (**a**) The cholesterol efflux was measured using the Cholesterol Efflux Assay kit. After treating the cells with Aβ_1-42_ oligomers (1 μM or 10 μM) for 24 h, two different cholesterol acceptors, ACM or apoA-I nanodiscs (40 μg/mL), were added. The fluorescence of supernatant and cell lysates was measured by spectrofluorometer. (**b**) Detection of cell viability, GFAP, and ABCA1 levels to confirm the apoA-I efficacy in restoring some cellular processes related to AD that were compromised by Aβ oligomers; * *p* <0.05; ** *p* ≤ 0.01; *** *p* ≤ 0.001 (one-way ANOVA).

**Figure 10 ijms-23-00102-f010:**
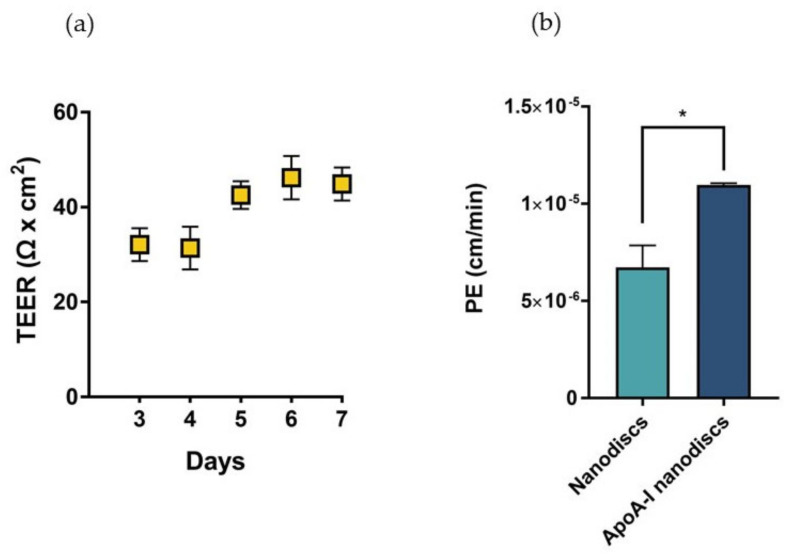
Characterization of the BBB in vitro model and evaluation of the passage of nanodiscs. (**a**) TEER values of the cell monolayer from day 3 to day 7 of growth on transwell reaching a maximum value of 44 Ω × cm^2^ on the seventh day of growth. (**b**) Evaluation of the passage of fluorescently labelled nanodiscs and apoA-I nanodiscs; * *p* <0.05 (Student’s *t* test).

**Table 1 ijms-23-00102-t001:** Physico-chemical characterization of apoA-I nanodiscs by DLS.

	Nanodiscs	ApoA-I Nanodiscs
Composition	POPC:CHOL:PEG-GE 35:40:25 (% mol)	POPC:CHOL:APOA-I 100:10:1 (% mol)
Ø (nm)	77.0 ± 0.4	92.6 ± 0.2
PDI	0.2 ± 0.001	0.2 ± 0.002

POPC = 1-palmitoyl-2-oleoyl-sn-glycero-3-phosphocholine; CHOL = cholesterol; PEG-PE = 1,2-distearoyl-sn-glycero-3-phosphoethanolamine-N-[amino(polyethylene glycol)-2000]; ApoA-I = apolipoprotein A-I.

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
