# Peer review of "Reduced Levels of ABCA1 Transporter Are Responsible for the Cholesterol Efflux Impairment in β-Amyloid-Induced Reactive Astrocytes: Potential Rescue from Biomimetic HDLs"

_ijms, 2021, doi:10.3390/ijms23010102_

Round 1

Reviewer 1 Report

The article is devoted to an interesting subject and all experiments are described carefully and in detail. However, the authors must consider several issues as follows:

  1. Please include the explanations for the composition of nanodiscs as a footer in Table 1, as the abbreviations “POPC”, “Chol”, “PEG-PE” and “apoA-I” were used here for the first time in the manuscript.
  2. For a better understanding of their work, I would recommend the authors to add in the manuscript subchapters at chapter 2. Materials and Methods,  in accordance with what is presented further in chapter 4. Materials and Methods, e.g. 2.1. Assessment of Aβ aggregation, 2.2. Assessment of cell viability after exposure to Aβ, 2.3. Evaluation of Aβ-induced astrogliosis, 2.4. Evaluation of cholesterol efflux, 2.5. Evaluation of ABCA1 and ABCG1 levels in reactive NHA, 2.6. Characterization of biomimetic nanodiscs, 2.7. Evaluation of apoA-I nanodiscs as cholesterol acceptors 2.8. Assessment of biomimetic nanodiscs ability to cross the BBB.
  3. Line 129-130: please explain more detailed the statement: “the presence of Aβ1-42 fibrils causes a dose-dependent reduction of GFAP expression levels, which are halved at Aβ1-42 concentration ≥ 10 μM” which is documented in Figure 4, since here it does not seem to be so clear that this effect is the same at 48h for the concentrations of 10 μM and 20 μM.
  4. Please add in the legends of Figures 4 and 6 the differentiated explanations for a) and b) (as for the other figures) in order to be better understood.
  5. Line 194: Please insert in the manuscript the explanations for “DLS” and “TEM” acronyms.
  6. Line 225-228: authors stated that “Results demonstrated that apoA-I nanodiscs promote a 2-fold increase (p=0.0155) of the cholesterol efflux from NHA cells treated with 1 μM Aβ1-42 oligomers, confirming their efficacy in rescuing the cholesterol homeostasis. A similar trend was also evident for higher Aβ1-42 dose (Figure 9)”, but for Aβ1-42 oligomers 10 μM the effect was visibly lower compared to 1 μM dose. The authors should explain this observation.
  7. Line 242-244: authors stated that “The results showed that TEER values progressively increased over time, reaching the maximum value of 44 Ω×cm2 on the 7th day after cell seeding (Figure 10a)”, but in Fig. 10a the higher value seems to be rather on the 6th day. The authors should explain this issue.

Reviewer 2 Report

This study aims to investigate how the Aβ1-42-induced astrogliosis affects the metabolism of cholesterol. Although the results are of interest, the following comments should be addressed for the support of the findings

1) Page 4, lines 120-123: Figure 3 _ If oligomers have become fibrils after 48h then we shouldn't expect the lowest NHA viability only when treated with the highest dose of oligomers for 48 h. Please explain this discrepancy

2) Figure 6: There is not good correlation of WB figures and graphs. A) Based on WB figures, the expression of ABCA1 is increased following treatment with 10uM Ab for 24h. B) Based on WB figures ABCG1 expression seems to be unaffected with 1uM Ab at 24h, to increase with 10uM Ab at 24h, to decrease with 10uM Ab at 48h and also to decrease with both 1 and 10uM Ab at 72h.

3) Figure 9: Is the result for 10uM Ab statistically significant?

4) Page 10, lines 275-277: The conclusions are not supported by the results.The authors could check weather apoA-I nanodisks also restore ABCA1 levels. In addition, they should check whether GFAP levels are restored. Furthermore, the effect of nanodisks in cell viability, in the presence of Ab, should be tested. As such, the findings do not support the beneficial effect of apoA-I nanodisks in AD. The authors should show that the restored cholesterol efflux in the presence of apoA-I nanodisks restores also a cellular process related to AD that was compromised by AB oligomers.

Round 2

Reviewer 2 Report

The authors have satisfactorily addressed my comments.